# Accuracy of Kinovea software in estimating body segment movements during falls captured on standard video: Effects of fall direction, camera perspective and video calibration technique

Nataliya Shishov[1]*, Karam Elabd[2], Vicki Komisar[1,3], Helen Chong[1], Stephen N. Robinovitch[1,2]

1 Department of Biomedical Physiology and Kinesiology, Simon Fraser University, Burnaby, British Columbia, Canada, 2 School of Engineering Science, Simon Fraser University, Burnaby, British Columbia, Canada, 3 School of Engineering, The University of British Columbia, Kelowna, British Columbia, Canada

* nshishov@sfu.ca

## Abstract

Falls are a major cause of unintentional injuries. Understanding the movements of the body during falls is important to the design of fall prevention and management strategies, including exercise programs, mobility aids, fall detectors, protective gear, and safer environments. Video footage of real-life falls is increasingly available, and may be used with digitization software to extract kinematic features of falls. We examined the validity of this approach by conducting laboratory falling experiments, and comparing linear and angular positions and velocities measured from 3D motion capture to estimates from Kinovea 2D digitization software based on standard surveillance video (30 Hz, 640x480 pixels). We also examined how Kinovea accuracy depended on fall direction, camera angle, filtering cut-off frequency, and calibration technique. For a camera oriented perpendicular to the plane of the fall (90 degrees), Kinovea position data filtered at 10 Hz, and video calibration using a 2D grid, mean root mean square errors were 0.050 m or 9% of the signal amplitude and 0.22 m/s (7%) for vertical position and velocity, and 0.035 m (6%) and 0.16 m/s (7%) for horizontal position and velocity. Errors in angular measures averaged over 2-fold higher in sideways than forward or backward falls, due to out-of-plane movement of the knees and elbows. Errors in horizontal velocity were 2.5-fold higher for a 30 than 90 degree camera angle, and 1.6-fold higher for calibration using participants' height (1D) instead of a 2D grid. When compared to 10 Hz, filtering at 3 Hz caused velocity errors to increase 1.4-fold. Our results demonstrate that Kinovea can be applied to 30 Hz video to measure linear positions and velocities to within 9% accuracy. Lower accuracy was observed for angular kinematics of the upper and lower limb in sideways falls, and for horizontal measures from 30 degree cameras or 1D height-based calibration.

**Data Availability Statement:** All relevant data are within the manuscript and its Supporting Information files.

**Funding:** This work was supported by operating grants from the Canadian Institutes of Health Research (funding reference numbers AMG-100487, TIR-103945, and TEI-138295) and the AGE-WELL National Centre for Excellence (AW CRP 2015-WP5.2, AWCRP-2020-04). NS was supported by a Simon Fraser University Graduate Dean's Entrance Scholarship and a CIHR Community Support Travel Award (164472). KE was supported by an AGE-WELL Network of Centres of Excellence in Technology and Aging Graduate Student Award. VK was supported by a Michael Smith Foundation for Health Research (MSFHR) Postdoctoral Award (18481), an AGE-WELL Network of Centres of Excellence in Technology and Aging Postdoctoral Award, and a CIHR Community Support Travel Award (164465). The funders had no role in study design, data collection and analysis, decision to publish, or preparation of the manuscript.

**Competing interests:** The authors have declared that no competing interests exist.

# 1. Introduction

Falls are the number one cause of unintentional injury and the number two cause of injury-related deaths world-wide [1,2]. Each year, falls cause 37 million hospital visits, 646,000 fatalities [2], and treatment costs over USD 50 billion globally [3]. Falls cause up to 80% of traumatic brain injuries [4] and 95% of hip fractures [5,6] in older adults. The high prevalence and clinical consequences of falls stress the need for improvements in the design and implementation of strategies for predicting, detecting, and preventing falls and injuries.

Improved understanding of the kinematics of real-life falls should help to inform efforts for injury prevention. For example, information on the movements of the body during falls can help exercise therapists in designing exercise programs for fall prevention [7–10], engineers in creating mobility aids [11–13] and architects and designers in creating safer environments [14–17]. Data on the kinematics of falls can also guide the design of improved sensor-based automatic fall detection systems [18–23]. Furthermore, data on the impact velocities of the body during falls is essential for the design and evaluation of fall protective gear [24–28] and compliant flooring.

Laboratory studies have used 3D motion capture to measure the kinematics of falls in young adults [29–32]. Studies have also examined the accuracy of wearable sensor systems and video-based approaches for detecting falls in the lab environment [21,33,34]. However, the kinematics of real-life falls may be considerably different from those observed in laboratory studies due to differences in environmental and situational factors (including the cause of the fall), and the population of interest.

2D planar video footage of real-life falls is increasingly available [35–38]. Previous studies have found that the frequency content of body movements during falls ranges from 1–10 Hz [32,39,40], which is below the 30 Hz (or frames per second) capture rate common to video surveillance cameras. However, falls often involve complex 3D movements, and validated approaches are required for extracting body segment kinematics from 2D video images. Furthermore, we need to understand how measurement accuracy depends on frame rate, camera orientation, and calibration technique [41]. Among the software packages available for estimating kinematics from planar video, there is growing use of the open-source software package Kinovea (http://www.kinovea.org). When compared to 3D motion capture, Kinovea estimated hip, knee, and ankle angles in the sagittal plane during walking with errors less than 5 degrees [42], and sagittal-plane knee angles during drop jumps with less than 10% error [43].

In the current study, we examined the accuracy of Kinovea for estimating the kinematics of falls from planar video with a resolution (640 x 480 pixels) and frame rate (30 Hz) typical of that acquired by surveillance video cameras. We addressed four objectives. First, we sought to determine the accuracy of Kinovea estimates under ideal video recording conditions, involving the camera axis oriented perpendicular to the plane of the fall [44], and calibration of the video (in m/pixel) based on a 2D grid recorded in the plane of the fall. We measured accuracy by comparing estimates from Kinovea to measures from high speed, 3D motion capture of the linear and angular positions and velocities of the head, torso, pelvis, upper limbs, and lower limbs. Second, we sought to determine how the accuracy of Kinovea was affected by the direction of the fall, and confirm whether our results support previous observations of good accuracy in kinematic estimates from planar video for backward falls [40], and higher errors for sideways falls, due to out-of-plane movements of the body segments [41]. Third, we examined how the accuracy of Kinovea is affected by the camera angle and frame rate (simulated through low-pass filtering the position data), which may vary widely in real-life conditions [35]. Fourth, we examined the accuracy of a simple 1D calibration approach based on the height of the faller, as a practical alternative to a 2D calibration grid in the plane of the fall. Our results provide a

basis for understanding what can be accurately measured, and for recommended approaches for using Kinovea to analyze the kinematics of falls from 2D planar video.

## 2. Methods

### 2.1 Participants

Thirty six falls were performed by three young adults who participated in this study (two women aged 32 and 22 years, and one man aged 34 years). None of the participants had a recent history of muscle strain, joint sprain, bone fracture, or concussion. The study was approved by the Office of Research Ethics at Simon Fraser University, and participants provided written informed consent prior to participating. Furthermore, all individuals whose image is shown in Fig 1 have given written informed consent (as outlined in PLOS consent form) to publish these images.

### 2.2 Protocol for falling experiments

Our protocol was designed to evaluate the accuracy of Kinovea for various fall directions and causes of imbalance. During the experiment, participants stood on a 4.4 x 3.8 m custom-built platform (Fig 1) that could rapidly translate in different directions in the horizontal plane via linear motors (Trilogy System Corporation, Webster, Texas, USA). The surface of the platform was covered by gym mats (30 cm thick) to prevent injuries during falls.

One set of trials was designed to simulate falls due to slipping or tripping. In these trials, falls were induced by suddenly translating the platform forward, backward, or sideways. The perturbation involved acceleration at 10 m/s$^2$ to a velocity of 2.2 m/s, followed by a period of constant velocity, and finally deceleration, over a total duration of 0.6 s [29]. A second set of trials was designed to simulate falls due to incorrect weight shifting (e.g., often characterized by excessive sway) [35]. In these trials, falls were self-initiated by the participant. In all trials, participants were instructed to "fall naturally," and wore a securely fitted helmet (Aspect, Smith (Sun Valley, USA)). Our analysis includes a total of 36 falls (six perturbation-based and six self-initiated falls in each of the forward, backward, and sideways directions).

### 2.3 Raw kinematic data from video (Kinovea) and motion capture (Qualisys)

During each trial, body movements were recorded for Kinovea analysis with four planar video cameras (Lorex LNZ44P4B, Markham, Canada), at a frame rate of 30 Hz and resolution of 640 x 480 pixels, selected to match common features of video from security cameras and sporting events. The Lorex cameras were positioned with their axes perpendicular to the plane of the fall (anterior and posterior to the participant during sideways falls, and to the left and right during forward and backward falls), and at angles of 30 and 60 degree to the plane of the fall (Fig 1). Using Kinovea (version 0.8.27), we manually digitized video footage from each camera angle frame-by-frame to determine vertical and horizontal marker positions from 10 frames before initiation of the fall to 10 frames after landing from the fall (cessation of movement). We refer to corresponding kinematic outcomes as "Kinovea" data.

We also recorded the time-varying, 3D positions of body segments with an 8-camera, passive-marker motion capture system at 600 Hz (Qualisys MIQUS, Gothenburg, Sweden). Each camera had a resolution of 6 megapixels, and the system was calibrated before each session. We refer to kinematic data from 3D motion capture as "Qualisys." For forward and backward falls reflective markers were placed (Fig 1A and 1B) on the participants' head (lateral aspects of helmet), shoulders (lateral to the glenohumeral joint), lateral humeral epicondyles, radial

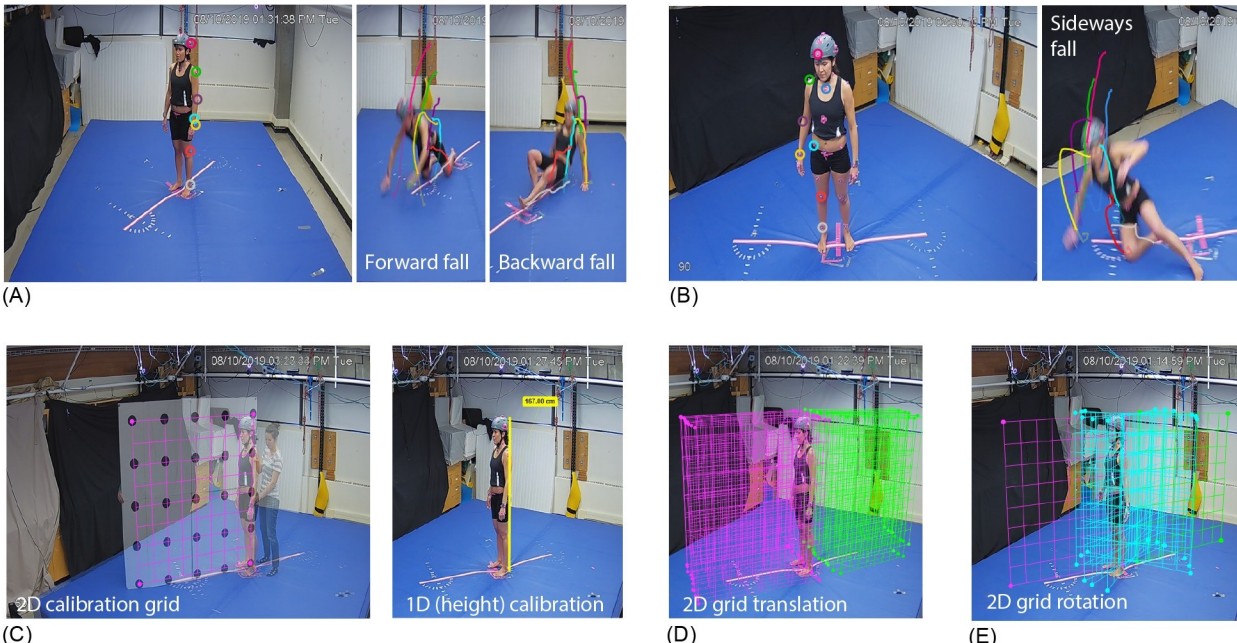

**Fig 1. Experimental setup.** Experimental setup for (A) forward and backward and (B) sideways falls illustrating Kinovea traces from digitization of key anatomical locations. Right panels display digitized traces from Kinovea of the time-varying positions during the falls of the head (pink), shoulder (green), elbow (purple), wrist (yellow), hip (light blue), knee (red), ankle (grey) and sternum (dark blue). Pink tape of the mat displays the plane of the fall. We examined how the accuracy of kinematic outcomes from Kinovea depended on (C) calibration technique (2D calibration grid versus 1D calibration based on participant height), (D) calibration frame translation (purple grids for forward falls and green grids for backward falls), (E) calibration grid rotation (blue grids), and camera view with respect to the fall (30, 60 and 90 degree views shown in (A), (C), and (B), respectively).

styloid processes, greater trochanters (GT), lateral femoral condyles, and lateral malleoli. For sideways falls, reflective markers were placed on participants' head (front of helmet), manubrium, acromion processes, lateral humeral epicondyles, radial styloid processes, anterior superior iliac spines (ASIS), midpoint of femoral condyles, and midpoint of malleoli. Pink tape was positioned around the markers to increase marker visibility in the videos.

Raw unfiltered position data from both Kinovea and Qualisys are provided as supporting information (S2–S5 Tables).

## 2.4 Data analysis

**2.4.1 Identification of Qualisys "ground truth" signals.** To arrive at a "ground truth" set of kinematic data from Qualisys (for comparison with other signals), we identified cut-off frequencies for low-pass filtering of Qualisys position data that removed non-physiological high frequency noise. From Fourier analysis, we found that less than 4% of the signal energy content was above 20 Hz, for 96.7% of all unfiltered vertical velocity traces (n = 365). Accordingly, Qualisys ground truth signals were based on position data processed with a 4th order, dual-pass Butterworth low-pass filter using a 20 Hz cut-off frequency. Visual inspection confirmed that the 20 Hz low-pass filter preserved the physiological components of the movement (Fig 2A–2C).

**2.4.2 Processing of kinematic data.** Kinovea data were converted from pixels to mm using the built-in calibration features of Kinovea software. The data were then low-pass filtered with a dual-pass 4th-order Butterworth filter, upsampled to 600 Hz, and temporally synchronized with Qualisys ground truth signals based on cross-correlation of head vertical velocity

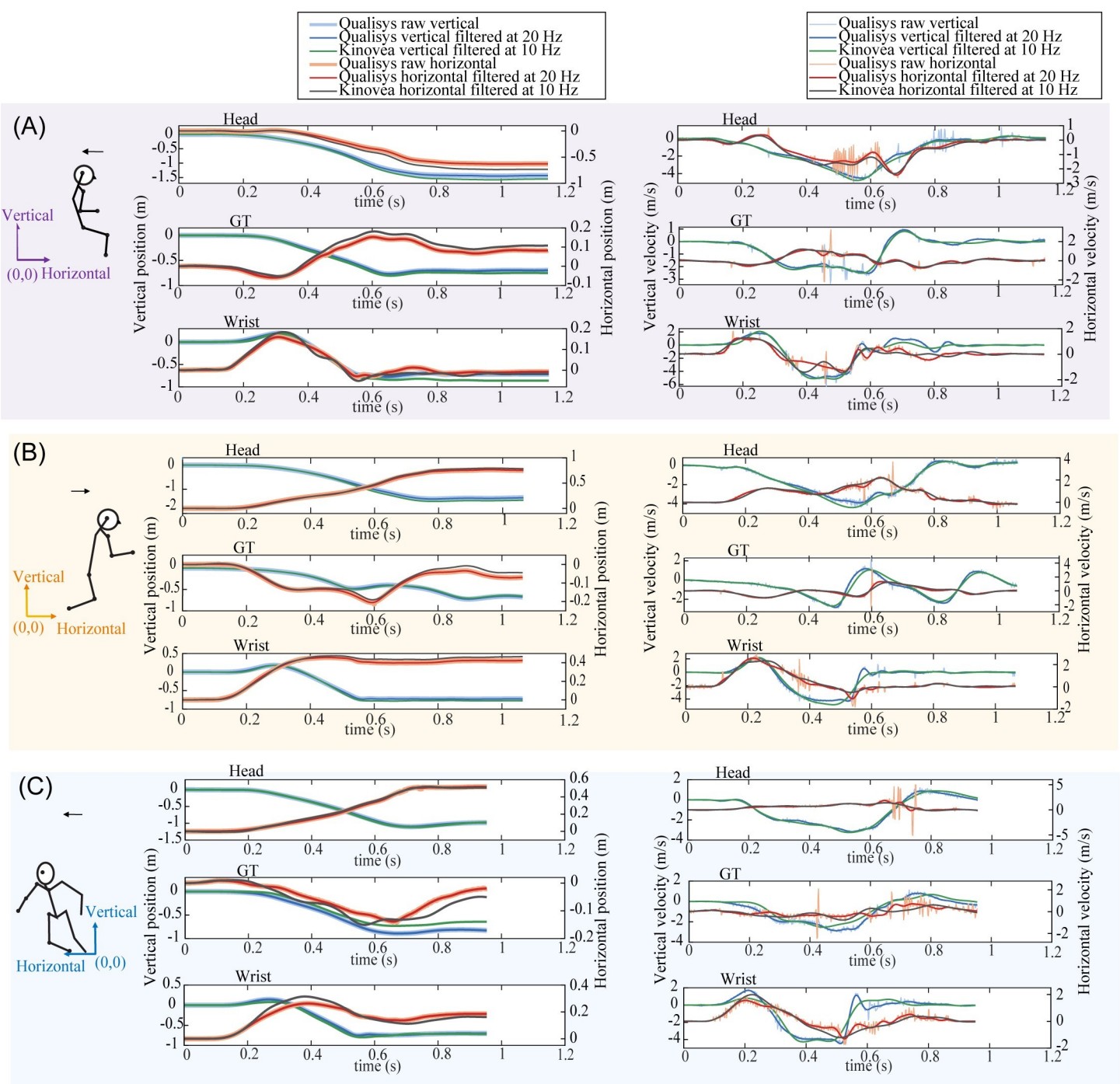

**Fig 2. Time-varying linear positions and velocities measured from 3D motion capture (Qualisys) and Kinovea.** The images show perturbation-based falls experienced by a 32 year old woman in the (A) backward fall direction, (B) forward fall direction, and (C) sideways fall direction. All traces are between onset of platform movement (time = 0) and end of fall (cessation of movement). Raw unfiltered data show considerable non-physiological noise. A 20 Hz low pass filter of Qualisys position data retained 96% of the energy content for 96.7% of the vertical velocity traces.

signals. We systematically examined how the accuracy of Kinovea signals depended on the filter cut-off frequency, and on the technique used in calibrating the videos (see section 2.4.5 below).

For both Qualisys and Kinovea, initial values were subtracted from horizontal (X axis) and vertical (Z axis) position data, to account for differences in the origins of the coordinate systems. Absolute angles of body segments were calculated from position data using the inverse tangent function. We focused on segment angles; rather than relative (joint) angles, since falls researchers are often interested in the orientation of body segments with respect to the vertical or the ground [45,46]. Head angles were calculated from head and shoulder markers during forward and backward falls, and head and sternum markers during sideways falls. Angles were calculated for the torso from shoulder and GT or ASIS, for the thigh from GT or ASIS and knee, for the upper arm from elbow and shoulder, for the forearm from elbow and wrist, and for the shank from knee and ankle markers. Linear and angular velocities were determined from numerical differentiation of position data using a first order central difference approach. Peak velocities (between fall initiation and end of fall) were identified for each anatomical location of interest.

**2.4.3 Accuracy measures.** We measured signal accuracy by comparing Qualisys ground truth data to either data derived from Kinovea, or data derived from alternative processing of Qualisys data (see section 2.4.6 below). Our measures of accuracy were: (a) the Root Mean Square Error (RMSE) in signals between the onset and end of the fall; (b) the RMSE normalized to the signal amplitude (NRMSE); and (c) the raw and absolute differences in peak velocities.

**2.4.4 Inter-rater reliability of Kinovea data.** We examined the inter-rater reliability of kinematic signals from Kinovea by comparing results from two raters, who independently digitized and calibrated videos of whole-body movements from three perturbation-based falls. RMSEs in linear position and velocity throughout the fall were compared for all camera angles, along with Intraclass Correlation Coefficients (ICC) (SPSS for Windows, version 25, IBM Corp., Armonk, N.Y., USA).

The mean RMSE between two raters was 0.011 m (SE = 0.001) for position, or 2.4% (SE = 0.2) of the signal amplitude, and 0.13 m/s (SE = 0.01) for velocity, or 4.8% (SE = 0.3) of the signal amplitude. ICC values and 95[th] confidence intervals were 1.00 (1.00–1.00) for linear position and 0.985 (0.984–0.986) for linear velocity.

**2.4.5 Effect of cut-off frequency and calibration technique on accuracy of Kinovea data.** We examined the accuracy of Kinovea data for cut-off frequencies of 3, 5, 7, 10, 12, and 14 Hz for low-pass filtering of position data. 14 Hz was selected as the maximum cut-off frequency for filtering the 30 Hz data, based on Nyquist considerations. We also examined how the accuracy of Kinovea outcomes differed between two approaches available in Kinovea for calibrating the videos (i.e., converting from pixels to m): a 2D grid versus a 1D vertical line (Fig 1C). The 2D grid was of dimensions 1.6 x 1.6 m, and contained a 5 x 5 array of dots spaced 40 cm apart. At baseline, the grid was located in the plane of the fall. The 1D vertical line was placed in a frame when the participant was standing prior to the onset of the fall. The line extended from the base of the foot to the top of the head, and was set equal to participant height. We also examined how signal accuracy was affected by: (a) translating the grid by ±10, ±20, ±30, ±40, and ±50 cm from the plane of the fall, where negative and positive values reflect translation away from and towards the camera, respectively (Fig 1D); (b) rotating the grid ±15, ±30, and ±45 degrees from the plane of the fall (Fig 1E); and (c) introducing errors in participant height of ±10 cm from the actual values.

**2.4.6 Qualysis "lower bound" signals.** We examined the bandwidth required for accurately capturing body movements during falls by comparing Qualisys ground truth data to Qualisys signals low pass filtered at lower cut-off frequencies. When compared to ground truth data, Qualisys signals filtered at 10 Hz had NRMSE less than 3% for horizontal and vertical position and velocity. Errors increased to 4% at 7 Hz, 5% at 5 Hz, and 8% at 3 Hz (Table 1).

**Table 1. Errors between position and velocity signals from Qualisys and Kinovea from different cut-off frequencies.**

|  | Vertical position | | Horizontal position | | Vertical velocity | | | | Horizontal velocity | | | |
|---|---|---|---|---|---|---|---|---|---|---|---|---|
|  | RMSE (m) | NRMSE (%) | RMSE (m) | NRMSE (%) | RMSE (m/s) | NRMSE (%) | Raw difference in peak velocity (m/s) | Percent difference in peak velocity (%) | RMSE (m/s) | NRMSE (%) | Raw difference in peak velocity (m/s) | Percent difference in peak velocity (%) |
| Q20-K14 | 0.051 ±0.002 | 9.2±0.6 | 0.035 ±0.002 | 6.0±0.3 | 0.22 ±0.01 | 6.5±0.2 | -0.04±0.02 | -1.3±0.9 | 0.16 ±0.01 | 7.3±0.2 | 0.03±0.02 | 0.9±1.0 |
| Q20-K12 | 0.051 ±0.002 | 9.2±0.6 | 0.035 ±0.002 | 6.0±0.3 | 0.22 ±0.01 | 6.5±0.2 | -0.04±0.02 | -1.1±0.9 | 0.16 ±0.01 | 7.2±0.2 | 0.32±0.02 | 1.2±1.0 |
| Q20-K10 | 0.050 ±0.002 | 9.2±0.6 | 0.035 ±0.002 | 6.0±0.3 | 0.22 ±0.01 | 6.5±0.2 | -0.02±0.02 | -0.5±0.9 | 0.16 ±0.01 | 7.2±0.2 | 0.05±0.02 | 2.1±1.0 |
| Q20-K7 | 0.050 ±0.002 | 9.2±0.6 | 0.035 ±0.002 | 6.0±0.3 | 0.23 ±0.01 | 6.7±0.2 | 0.03±0.02 | 1.2±0.9 | 0.16 ±0.01 | 7.2±0.2 | 0.10±0.02 | 5.0±1.0 |
| Q20-K5 | 0.051 ±0.002 | 9.2±0.6 | 0.035 ±0.002 | 6.0±0.3 | 0.25 ±0.01 | 7.2±0.2 | 0.11±0.03 | 4.3±1.0 | 0.17 ±0.01 | 7.7±0.2 | 0.18±0.02 | 9.6±0.9 |
| Q20-K3 | 0.051 ±0.002 | 9.4±0.6 | 0.037 ±0.001 | 6.4±0.3 | 0.32 ±0.01 | 8.9±0.2 | 0.33±0.03 | 12.0±1.2 | 0.22 ±0.01 | 9.8±0.2 | 0.37±0.02 | 19.6±1.0 |
| Q20-Q14 | 0.001 ±0.001 | 0.1±0.1 | 0.001 ±0.001 | 0.1±0.1 | 0.03 ±0.01 | 0.7±0.1 | 0.02±0.01 | 0.8±0.1 | 0.02 ±0.01 | 1.0±0.1 | 0.02±0.01 | 1.1±0.1 |
| Q20-Q12 | 0.001 ±0.001 | 0.1±0.1 | 0.001 ±0.001 | 0.1±0.1 | 0.04 ±0.01 | 1.1±0.1 | 0.03±0.01 | 1.3±0.1 | 0.03 ±0.01 | 1.5±0.1 | 0.03±0.01 | 1.8±0.0 |
| Q20-Q10 | 0.001 ±0.001 | 0.2±0.1 | 0.001 ±0.001 | 0.1±0.1 | 0.06 ±0.01 | 1.6±0.1 | 0.06±0.01 | 2.3±0.2 | 0.05 ±0.01 | 2.1±0.1 | 0.06±0.01 | 3.0±0.2 |
| Q20-Q7 | 0.002 ±0.001 | 0.4±0.1 | 0.001 ±0.001 | 0.3±0.1 | 0.11 ±0.01 | 2.8±0.1 | 0.13±0.01 | 4.8±0.4 | 0.08 ±0.01 | 3.3±0.1 | 0.12±0.01 | 6.2±0.4 |
| Q20-Q5 | 0.004 ±0.001 | 0.8±0.1 | 0.003 ±0.001 | 0.6±0.1 | 0.16 ±0.01 | 4.3±0.1 | 0.22±0.01 | 8.3±0.5 | 0.11 ±0.01 | 4.8±0.1 | 0.21±0.01 | 10.7±0.5 |
| Q20-Q3 | 0.010 ±0.001 | 2.0±0.1 | 0.007 ±0.001 | 1.7±0.10 | 0.27 ±0.01 | 7.2±0.1 | 0.45±0.02 | 16.6±0.8 | 0.18 ±0.01 | 7.7±0.1 | 0.41±0.02 | 21.0±0.7 |

**NOTES:** Comparisons are between Qualisys ground truth signals (Q20) and either Kinovea signals (top six rows) or Qualisys signals (bottom six rows) from low-pass filtering of position data at different cut-off frequencies (e.g., 14 Hz for "K14" and "Q14"). Results show mean ± 1 SE values averaged across all body parts and fall directions. Results are specific to Kinovea data from the 90 degree camera angle, calibrated with the calibration frame located in the plane of the fall. For positions, values less than 0.001 m are rounded up. For velocities, values less than 0.01 m/s are rounded up. Percent differences (NRMSE and peak velocities) are calculated for each fall and then averaged across falls.

Peak vertical velocities matched ground truth values to within 0.06 m/s (3%) at 10 Hz, 0.13 m/s (5%) at 7 Hz, 0.22 m/s (9%) at 5 Hz, and 0.45 m/s (17%) at 3 Hz (Table 1 and Fig 2). Based on these findings, we regarded 10 Hz as an appropriate "lower bound" filter cut-off frequency that provides over 95% accuracy in capturing body segment positions and velocities during falls, and represents the theoretical upper limit on the accuracy of video cameras capturing at 10 Hz.

**2.4.7 Magnitude of out-of-plane movement during falls.** Finally, we analyzed Qualisys data to quantify peak displacements and velocities of body segments in the "out-of-plane" (Y) axis during falls, which could not be captured by our 2D Kinovea analysis. We also compared horizontal X positions and velocities to resultant XY signals from Qualysis.

## 2.5 Statistical analysis

We used one-way ANOVAs to examine the effect of the following methodological variables on the accuracy of Kinovea signals: Kinovea filter cut-off frequency, camera angle, fall direction,

body part, and calibration technique. Each fall was treated as an independent measure. Fall direction was analyzed with independent samples ANOVAs because the comparisons were between different falls. All other variables were analyzed with repeated measures ANOVAs (with the fall treated as a random factor) because these variables related to different approaches (or treatments) for collecting or processing data from the same set of falls. In case of significant main effects, Tukey's Post hoc analysis was used to test pair-wise comparisons. All ANOVA analyses were conducted using JMP software (Version 15. SAS Institute Inc., Cary, NC) using a significance level of $\alpha < 0.05$.

## 3. Results

Based on Qualisys ground truth signals, the head displaced by average vertical and horizontal distances of 1.109 and 0.976 m, rotated by 50.9 degrees, and experienced peak vertical and horizontal velocities of 3.04 and 2.26 m/s and peak angular velocities of 415 degree/s. Figs 2 and 3 show time-varying linear and angular positions and velocities for a sample backward, forward, and sideways fall.

Unless specified otherwise, the comparisons below are for Kinovea results from cameras oriented at 90 degree (camera in front during sideways falls and to the right during forward and backwards falls), the calibration grid placed in the plane of the fall, and low-pass filtering of position data with a 10 Hz cut-off frequency.

### 3.1 Effect of filtering on the accuracy of fall kinematics measured by Kinovea

Based on comparison to Qualisys ground truth signals, the accuracy of Kinovea signals depended on the cut-off frequency used in low-pass filtering Kinovea position data (Table 1 and Fig 4). There was a significant effect of cut-off frequency on mean RMSEs for horizontal (but not vertical) and angular position ($p \leq 0.0002$), and for vertical, horizontal, and angular velocity ($p < 0.0001$). Post hoc analysis showed that RSMEs for all outcomes were not different between 7, 10, 12, and 14 Hz. RSMEs for position were lower at 3 Hz ($p \leq 0.0120$), and for velocities at 5 Hz ($p \leq 0.0299$), than for higher cut-off frequencies. With a 10 Hz cut-off frequency, mean RMSEs were 0.050 m (SE = 0.002) or 9.2% of the total signal amplitude for vertical position, 0.036 m (SE = 0.002) or 6% for horizontal position, 0.22 m/s (SE = 0.01) or 6.5% for vertical velocity, 0.16 m/s (SE = 0.01) or 7.2% for horizontal velocity, 10.1 degree (SE = 0.9) or 13.0% for angular position, and 63 degree/s (SE = 4) or 10.0% for angular velocity (Table 1, Fig 4 and S1 Table).

### 3.2 Effect of fall direction and body part on the accuracy of fall kinematics measured by Kinovea

There was a significant effect of fall direction on the accuracy of Kinovea for all linear and angular measures ($p \leq 0.0043$; Fig 5). Post hoc analysis showed that all outcomes except horizontal positions had smaller RMSEs in forward than sideways falls ($p \leq 0.0035$). The differences were greatest for angular positions and velocities, where NRMSEs averaged 25.6 and 15.6% in sideways falls, and 6.7 and 7.2% for forward and backward falls combined. Angular positions, and vertical and angular velocities, were more accurate in backward than sideways falls ($p \leq 0.0005$). Linear positions but not velocities were less accurate in backward than forward falls ($p \leq 0.0122$).

For the 90 degree camera, $R^2$ values for the correlations between peak vertical velocities from Kinovea and Qualisys were 0.911 for backward falls, 0.891 for forward falls, and 0.864 for

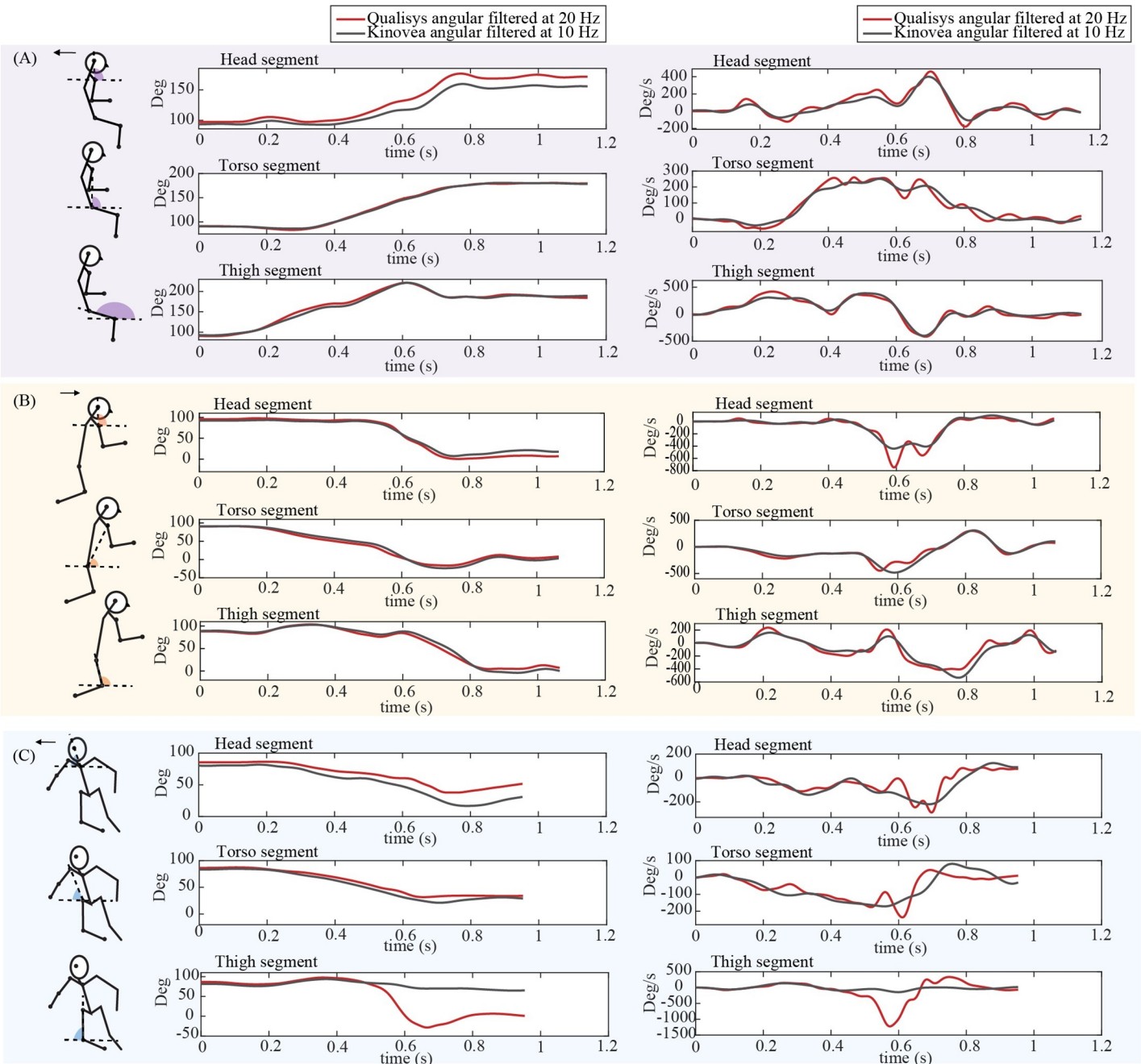

**Fig 3. Angular positions and velocities measured from 3D motion capture (Qualisys) and Kinovea.** The traces show time-varying angular positions (with respect to the horizontal) and velocities for selected body segments, measured for (A) backward, (B) forward, and (C) sideways perturbation-based falls. Data are for the same falls as shown in Fig 2.

sideways falls (Fig 6). The slope of the lines varied between 0.83 and 1.09. The mean signed error averaged 2% or lower for both peak vertical velocity and peak horizontal velocity for the 90 degree camera, and was different than zero only for forward and sideways falls from the 30 degree camera, reflecting no bias in Kinovea estimates from the 60 and 90 degree cameras.

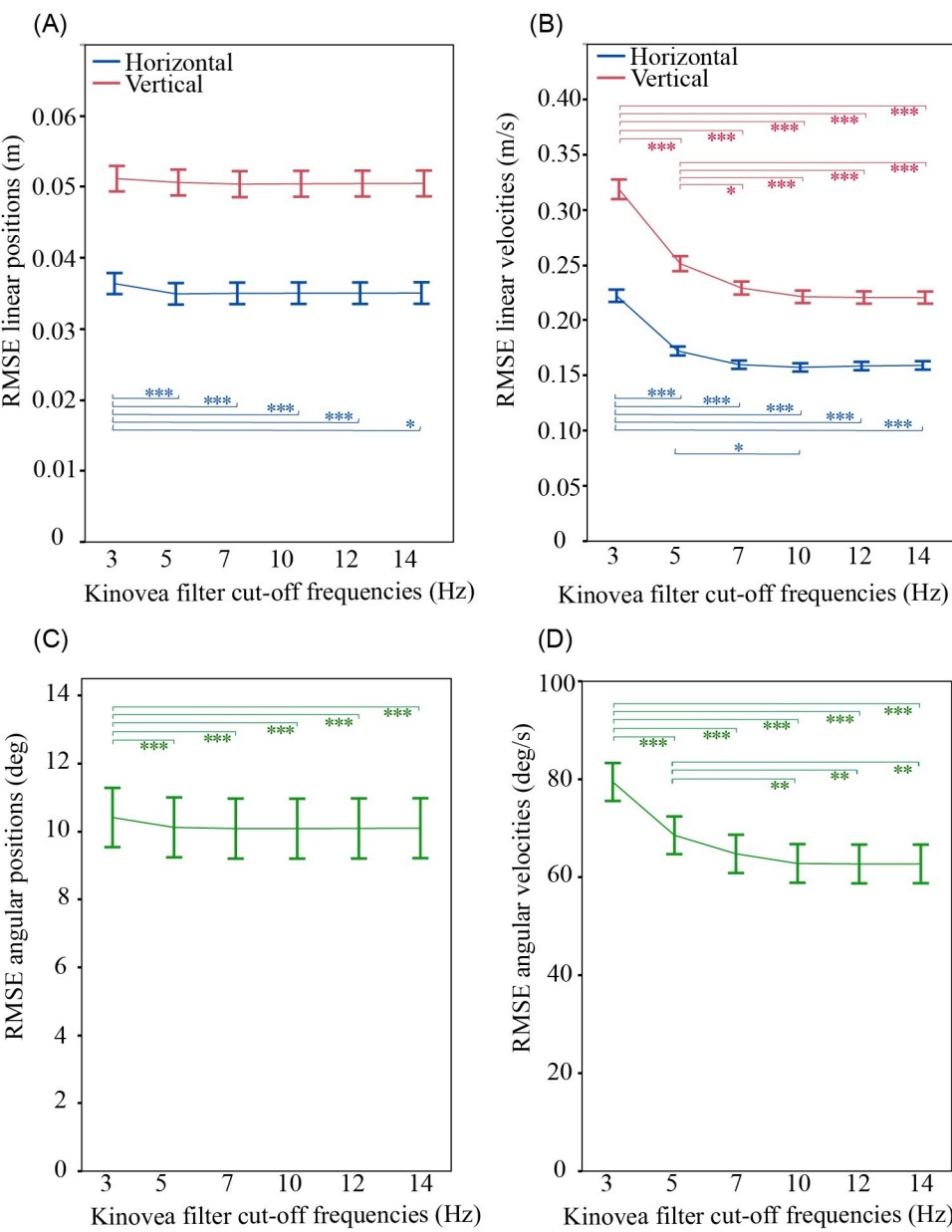

**Fig 4. Effect of low-pass filter cut-off frequency on the accuracy of Kinovea estimates.** (A) Linear position, (B) linear velocity, (C) angular position, and (D) angular velocity. The plots show root mean square errors (RMSE) for Kinovea data filtered at cut-off frequencies of 3, 5, 7, 10, 12 and 14 Hz, based on comparison with Qualisys ground truth signals. Results are specific to Kinovea data from the 90 deg camera angle, calibrated with the calibration frame located in the plane of the fall. Data points show mean values averaged across all body parts and fall directions with error bars showing ± 1 SE. * p≤0.05, ** p≤0.01, *** p≤0.001.

### 3.3 Effect of body part on the accuracy of fall kinematics measured by Kinovea

Body parts had a significant effect on mean RMSE for all outcome measures (p<0.0001; Fig 7). For linear measures, the elbow and wrist had at least 20% higher mean RMSEs than other body parts (except for vertical velocities, where the RMSE for ankle was similarly high). For angular measures, the thigh and forearm had at least 42% higher mean RMSEs than other body segments, driven by large errors in sideways falls.

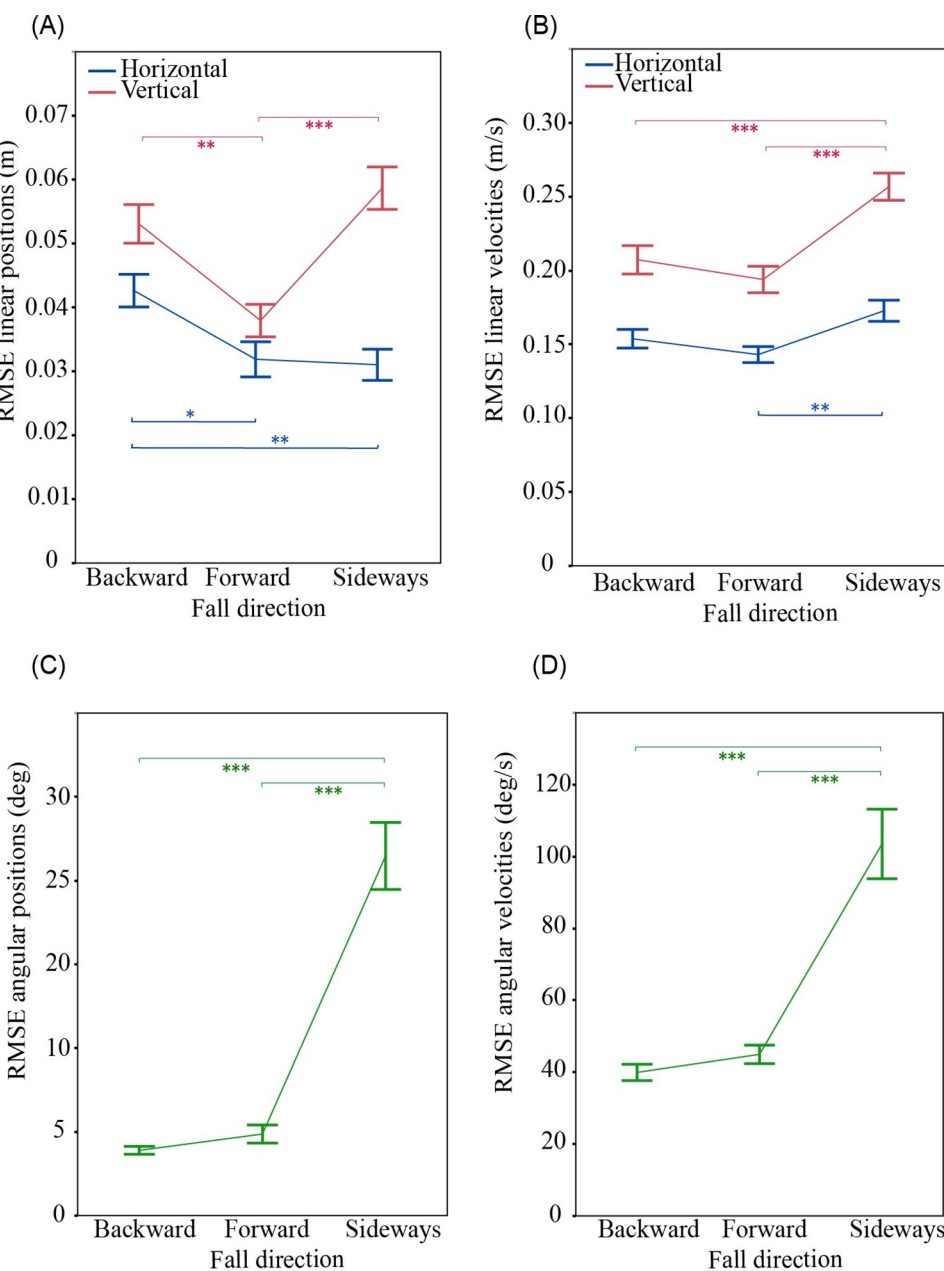

**Fig 5. Effect of fall direction on accuracy of Kinovea estimates.** (A) Linear position, (B) linear velocity, (C) angular position, and (D) angular velocity. RMSEs are based on comparison of Qualisys ground truth signal to Kinovea data from the 90 deg camera angle, calibrated with the calibration frame located in the plane of the fall, with position data filtered at 10 Hz. Data points show mean values averaged across all body parts with error bars showing ± 1 SE. * p≤0.05, ** p≤0.01, *** p≤0.001.

## 3.4 Effect of camera angle on the accuracy of fall kinematics measured by Kinovea

There was a significant effect of camera angle on mean RMSE for all kinematic outcomes (p≤0.0004; Figs 6 and 8). Post hoc analyses showed significantly higher RMSEs for the 30 degree than 90 and 60 degree camera angles for all outcomes (p≤0.0288). At 30 degree, the RMSE values for horizontal and angular measures were over two-fold greater than for 90

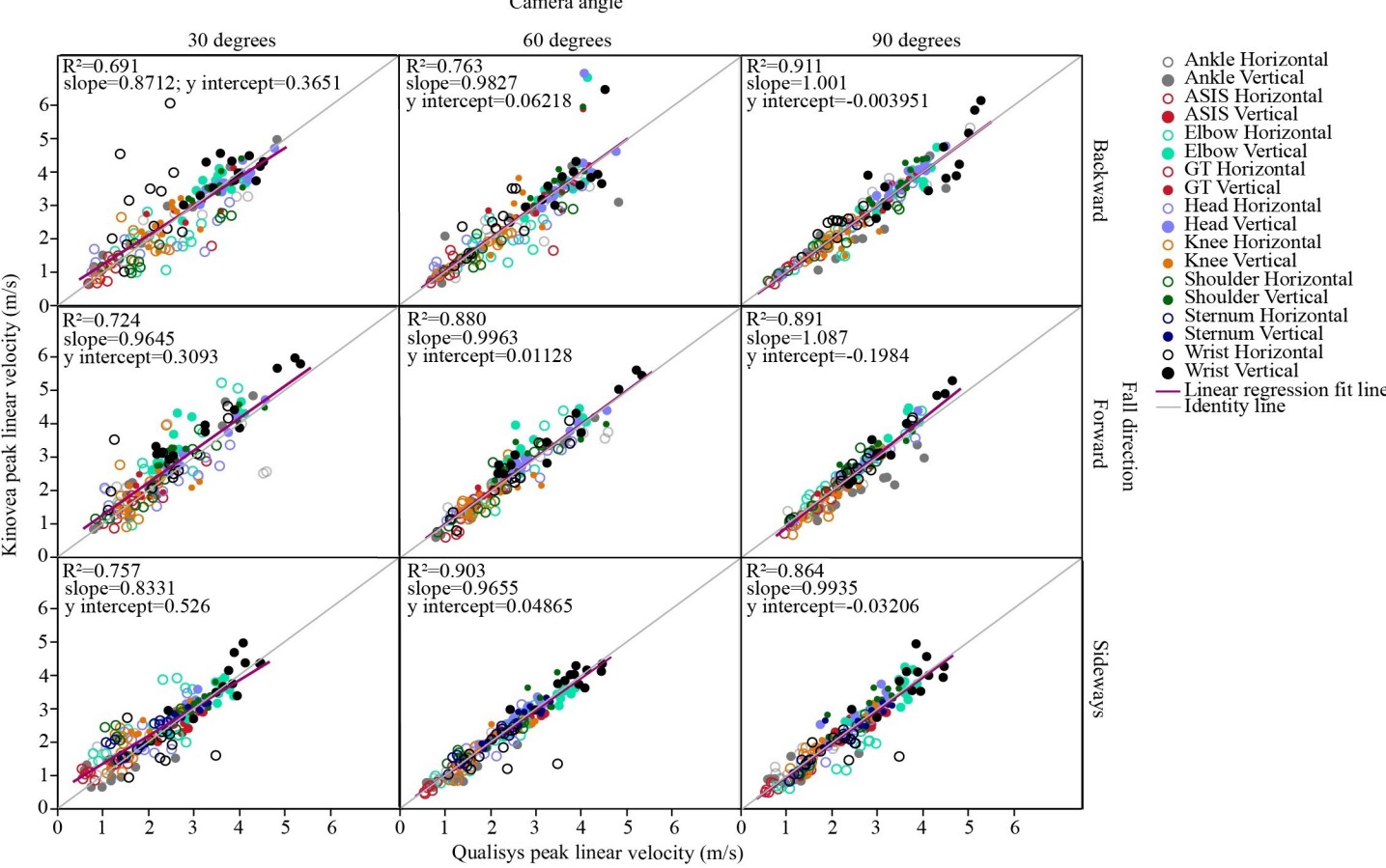

**Fig 6. Effects of camera angle and fall direction on agreement between Kinovea and Qualisys peak velocities.** The graphs compare peak vertical and horizontal velocities for various anatomical locations. Results are specific to the calibration grid located in the plane of the fall and Kinovea velocities differentiated from position data filtered with a cut-off frequency of 10 Hz.

degree. There were differences only for horizontal velocities (p = 0.0076) in RMSEs between 60 than 90 degree camera angles.

## 3.5 Effect of calibration on the accuracy of fall kinematics measured by Kinovea

The accuracy of Kinovea outcomes depended on the orientation of the calibration grid (Fig 9). Both the direction and magnitude of grid translation associated with errors in linear positions and velocities (p<0.0001). Grid translation, averaged for negative and positive displacements (movement of the grid away and towards the camera, respectively), of 10 and 20 cm did not affect the accuracy (p≥0.0922). Average RMSE values for 30 and 50 cm translation were 10 and 24% greater than at baseline. For larger translations, mean RMSEs for linear outcomes increased more for negative than positive translations (p<0.0001). Furthermore, for grid translations in the positive direction there was no effect on accuracy of vertical and horizontal linear velocities for translations up to 40 cm (p≥0.0762) and 30 cm (p≥0.8186). Conversely, negative displacement of the grid beyond 10 cm resulted in reduced accuracy (p<0.0001). Distance of grid translation did not significantly effect mean RMSEs for angular positions (p = 0.0794) and velocities (p = 0.6852).

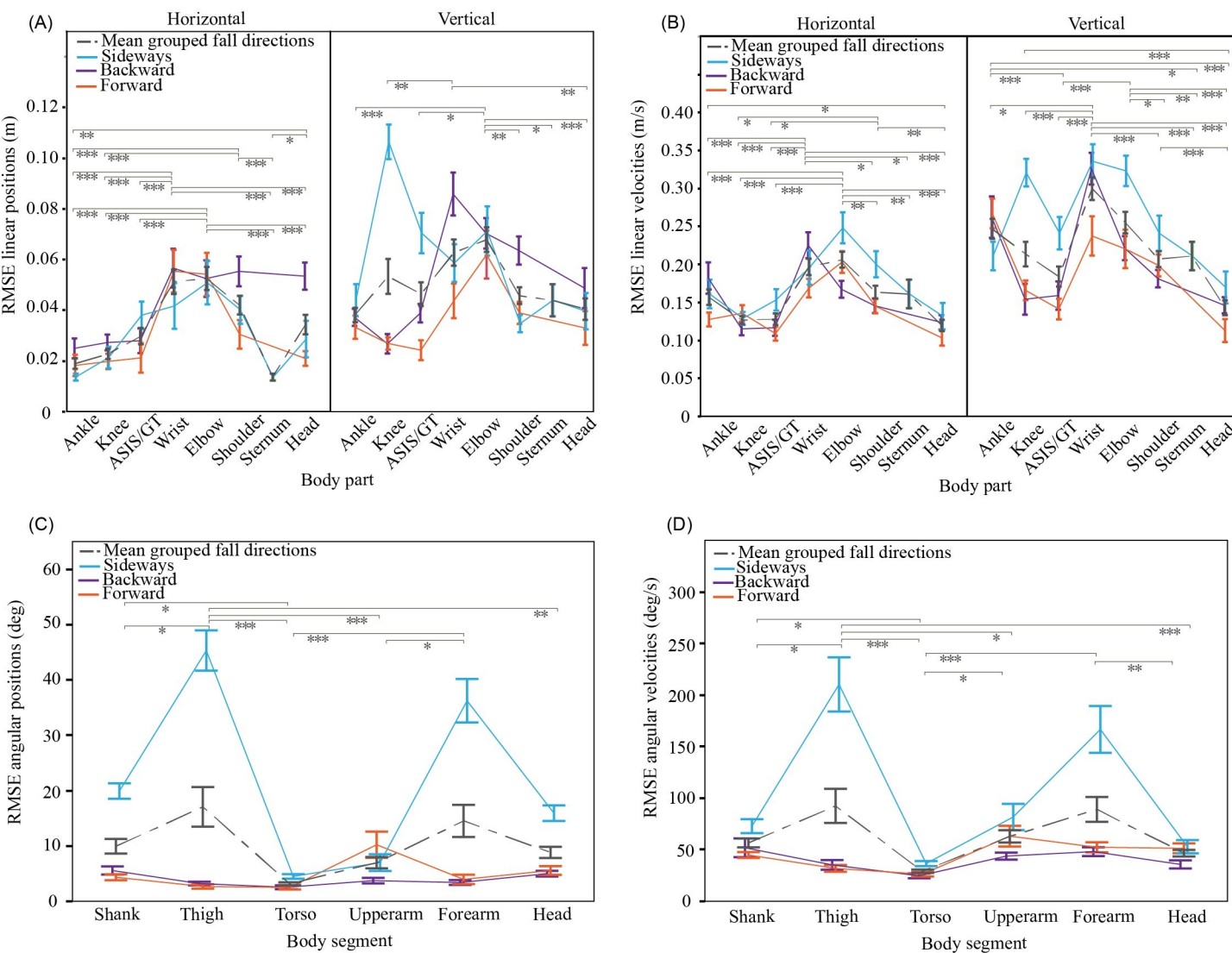

**Fig 7. Differences across body sites and fall directions in the accuracy of Kinovea estimates.** (A) Linear position, (B) linear velocity, (C) angular position, and (D) angular velocity. Root mean square errors (RMSE) are from comparison between Qualisys ground truth signals and Kinovea estimates based on the 90 deg camera angle, the calibration frame located in the plane of the fall, and filtering of position data with a 10 Hz cut-off frequency. Data points show mean values, and error bars show ± 1 SE (sideways = blue, backward = purple, forward = orange; grey = mean values across all fall directions). Significant differences are shown for mean values across all fall directions (* p≤0.05, ** p≤0.01, *** p≤0.001).

Rotation of the calibration grid also had a significant effect on the accuracy of all Kinovea outcomes (Fig 9). Mean RMSEs did not differ between 0 and 15 degree, except for vertical linear positions (p = 0.0375), but thereafter increased with grid rotation (p≤0.0185). Again, the effect was greater for linear than angular velocities. For linear velocities, RMSEs at 30 and 45 degree were ~2 and ~3-fold larger, respectively, than values for the grid oriented in the plane of the fall.

Mean RMSEs were higher for all outcomes when videos were calibrated based on participant height versus the calibration grid in the plane of the fall (p≤0.0031; Fig 9). The difference was more pronounced for horizontal velocities (where the mean NRMSE increased from 7.2% to 11.2%; S1 Table) than for vertical velocities (6.5% versus 6.8%). Introducing errors in the value of body height used in the video calibration affected the accuracy of linear (p<0.0001), but not angular measures (p≥0.7644; Fig 9).

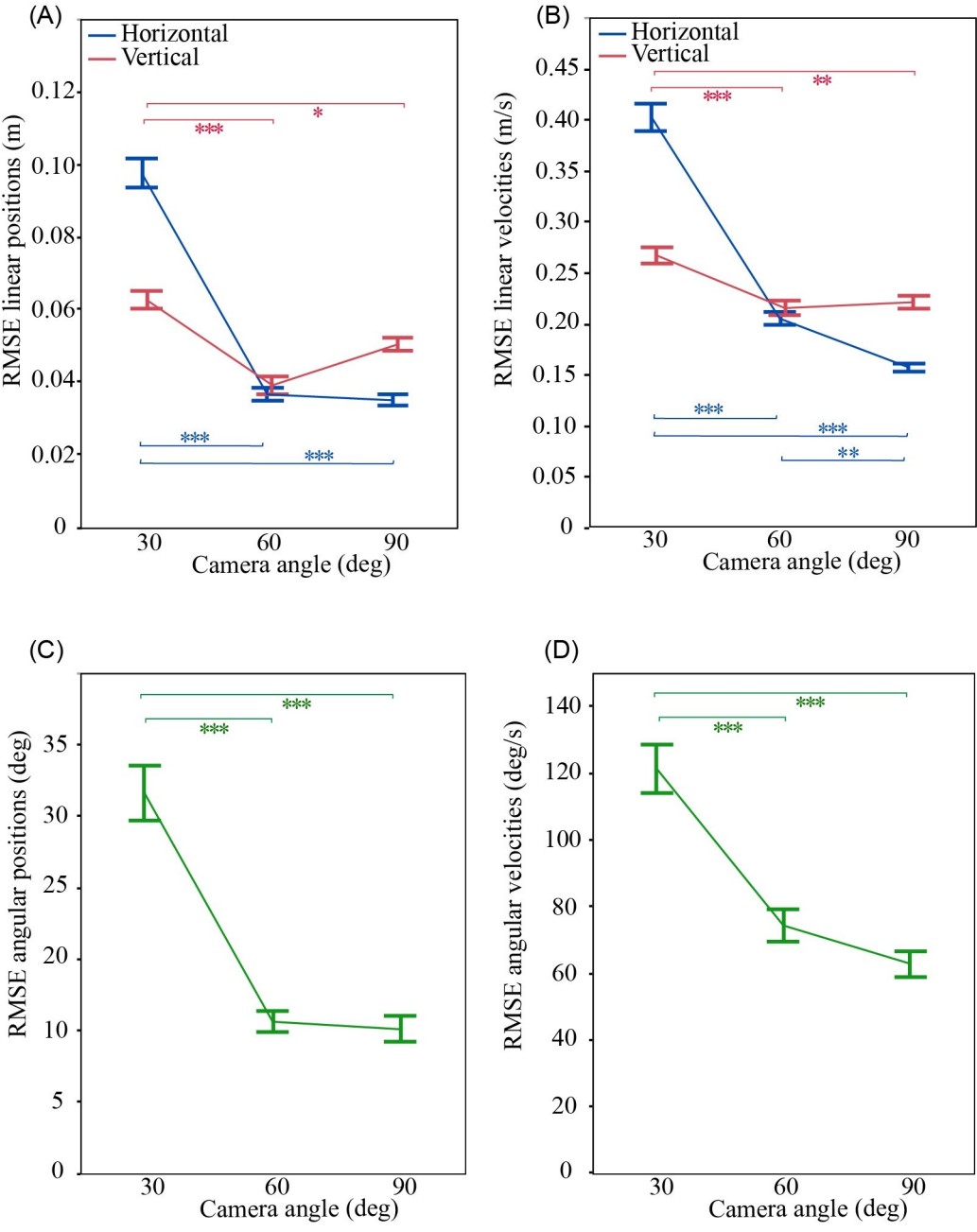

**Fig 8. Effect of camera angle on the accuracy of Kinovea estimates.** (A) Linear position, (B) linear velocity, (C) angular position, and (D) angular velocity. Root mean square errors (RMSE) are from comparison between Qualisys ground truth signals and Kinovea estimates based on the calibration frame located in the plane of the fall, and filtering of position data with a 10 Hz cut-off frequency. Data points show mean values, and error bars show ± 1 SE. * p≤0.05, ** p≤0.01, *** p≤0.001.

## 3.6 Magnitude of out-of-plane movement during falls measured by Qualisys

From Qualisys 3D motion capture data, the peak magnitudes of out-of-plane (Y) horizontal displacement and velocity, averaged over all body parts, were 0.089 (SD = 0.060), 0.099 (0.050), and 0.158 (0.066) m, and 0.73 (0.51), 0.79 (0.48), and 0.96 (0.60) m/s, during backward, forward and sideways falls, respectively. In comparison, peak in-plane (X) horizontal

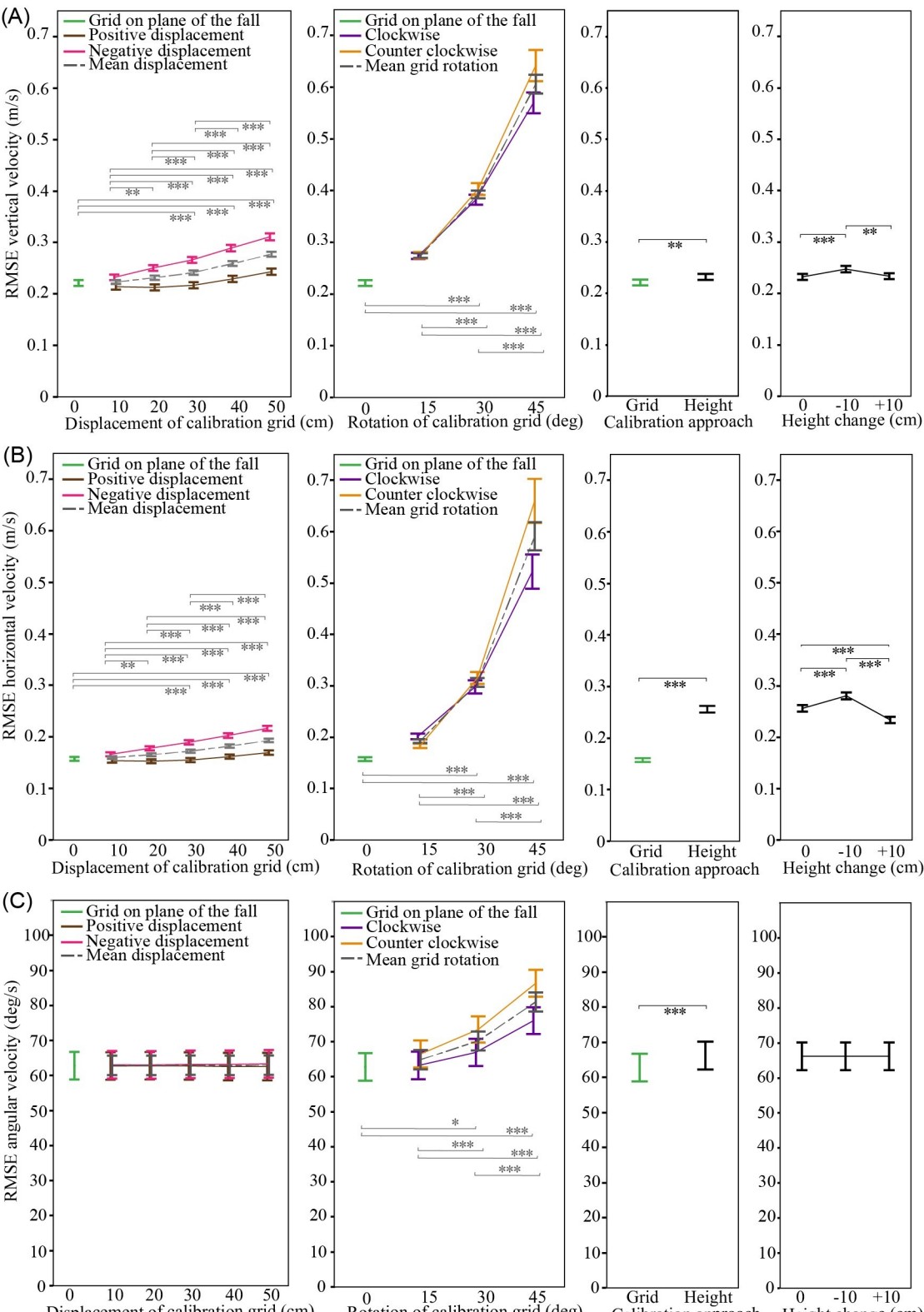

**Fig 9. Effect of calibration technique on the accuracy of Kinovea estimates.** (A) Vertical velocity, (B) horizontal velocity, and (C) angular velocity. Root mean square errors (RMSE) are from comparison between Qualisys ground truth signals and Kinovea estimates from the 90 deg camera angle, and filtering of position data with a 10 Hz cut-off frequency. Data points show mean

values, and error bars show ± 1 SE. Statistical comparisons in two left columns are based on mean values. * p≤0.05, ** p≤0.01, ***
p≤0.001.

displacement and velocity during backward, forward, and sideways falls averaged 0.664 (0.400), 0.810 (0.528), and 0.658 (0.420) m, and 2.04 (0.88), 2.13 (0.84), and 1.74 (0.68) m/s. Mean out-of-plane movements larger than 0.150 m were observed during backward falls only for the wrist (0.161 (0.084) m; mean peak out-of-plane velocity = 1.27 (0.56) m/s), and during sideways falls for the elbow (0.195 (0.061) m; 2.19 (0.60) m/s), ASIS (0.172 (0.051) m; 0.74 (0.18 m/s)), and knee (0.207 (0.026) m; 1.01 (0.47) m/s). Mean RMSEs between X (1D) and resultant XY (2D) horizontal positions and velocities during backward, forward and sideways falls were 0.010 (0.014), 0.009 (0.016), and 0.023 (0.030) m, and 0.10 (0.10), 0.10 (0.09), and 0.24 (0.30) m/s, respectively.

## 4. Discussion

We examined the accuracy of Kinovea software (when compared to high-speed, 3D motion capture) in estimating the kinematics of the body during falls from video footage collected by standard surveillance cameras (30 Hz, 640 x 480 resolution). We also examined how the accuracy of Kinovea depended on fall direction, camera angle, video calibration technique, and cut-off frequency for low-pass filtering of position data. Below, we provide guidance for those considering the use of Kinovea for analyzing the kinematics of video-captured falls. We discuss the accuracy of Kinovea observed under both "ideal" and non-ideal conditions, where ideal conditions involved: (a) the camera axis oriented perpendicular (90 degrees) to the plane of the fall, (b) the video calibrated from a 2D grid recorded in the plane of the fall, and (c) a 10 Hz cut-off frequency used for low-pass filtering of position data. We also highlight the conditions where the accuracy of specific parameters was substantially lower than values observed under ideal conditions, and discus the potential reasons for the observed trends.

We stress that, for a given user, the acceptability of the observed errors will depend on the user's required level of accuracy, which in turn depends on the study goals and design. Consider, for example, a situation where a user is interested in comparing peak vertical velocities in forward versus backward falls [41], or in falls experienced by women versus men [47]. For these applications, the small error of Kinovea in measuring peak vertical velocity over a wide range of conditions should allow the user to detect small differences between conditions (if they exist). In contrast, if the user instead wishes to compare the time-varying angular velocities of body segments [48], for which Kinovea had a substantially higher measurement error, the (actual) difference between conditions may need to be quite large, to overcome the inherent errors and be detected from Kinovea-based analysis. Supplementary S1 Table allows readers to identify the expected level of accuracy for specific combinations of kinematic outcome, body part, fall direction, and approach for video collection and calibration.

Under ideal conditions, Kinovea estimated vertical and horizontal positions and velocities over the duration of the fall to within 9% error, angular positions and velocities to within 13% error, and peak vertical and horizontal velocities to within 2% error and with no bias (or constant error). These baseline errors were likely due to human inaccuracy in digitization of the video footage with Kinovea, and perhaps the limited (640 x 480) resolution of the video. It is unlikely that the errors arose from systemic causes (since Kinovea was just as likely to overestimate as underestimate peak velocities), or from limitations related to the 30 Hz video frame rate. Our results agree with previous studies which found that the frequency content of body movements during falls is between 1–10 Hz [32,39], and therefore capable of being captured accurately from 30 Hz video. From Qualisys 3D motion capture at 600 Hz, we found that 96%

of the energy in kinematic signals was below 20 Hz, and a 10 Hz cut-off frequency for low-pass filtering of position data successfully filtered out non-physiological noise with minimal distortion of the underlying signal, even for rapidly-moving body parts such as the hands. For Kinovea, when compared to 10 Hz, filtering at 3 Hz caused velocity errors to increase 1.4-fold. However, the errors in all Kinovea outcomes did not change over cut-off frequencies between 7 and 14 Hz (the maximum value based on Nyquist considerations). While further research is required, our results suggest that data from 15 Hz cameras, filtered at 7 Hz, may provide a level of accuracy similar to the values we observed.

We found that the accuracy of Kinovea depended on fall direction. Since Kinovea involves a 2D analysis of planar video, its accuracy will be reduced by out-of-plane movements of body segments (orthogonal to the plane of the video image). From our Qualisys 3D measures, out-of-plane movements were largest at the knee and elbow in sideways falls, and at the wrist in backward falls. This explains why, when compared to other fall directions, Kinovea errors in sideways falls were about 4-fold higher for thigh and forearm angular position and velocity, and about 2-fold higher for knee and elbow vertical velocity. Furthermore, errors in wrist vertical velocity were about 2-fold higher in backward and sideways falls, than in forward falls. Users should be cautious in relying on Kinovea to yield accurate results for these particular combinations of fall direction and kinematic outcome.

Errors were more stable across fall directions for kinematics of the head, sternum, and pelvis. In particular, we found encouraging evidence for the accuracy of Kinovea in estimating pelvis impact velocities in sideways falls, which are relevant to the cause and prevention of hip fractures [49,50]. Unlike Choi et al. [41], who observed errors that they deemed as unacceptably high for peak pelvis velocities from digitizing planar video of sideways falls (mean error = 1.01 m/s (SD = 0.59) for vertical velocity, and 0.13 m/s (SD 0.54) for horizontal velocity), we found that Kinovea estimated peak velocities of the ASIS during sideways falls to within an error of 0.12 m/s (3.9%) for vertical velocity, and 0.14 m/s (10.7%) for horizontal velocity. It is difficult to explain why the errors observed by Choi for sideways falls were similar to ours for peak horizontal pelvis velocity, but much higher for peak vertical pelvis velocity, but this may relate to Choi et al. examining only a single fall in each of the forward, sideways and backward directions, while we analyzed 12 falls in each direction.

Since real-life falls may rarely occur in planes that are perpendicular to the axis of a nearby surveillance camera, we explored how the accuracy of Kinovea outcomes differed between 30, 60 and 90 degree camera angles. We found that errors in vertical kinematics were relatively unaffected by camera angle, with the 30 degree camera providing less than 12% error for vertical positions and 8% error for vertical velocities. This suggests that Kinovea is appropriate for estimating vertical kinematics for a wide range of camera angles. However, the camera angle more strongly affected the accuracy of Kinovea estimates of horizontal and angular kinematics, likely due to its effect on the range of observable horizontal and angular movement. Errors in horizontal velocity were 2.5-fold higher for a 30 than 90 degree camera angle. Differences were small between 60 and 90 degree camera angles, and limited to horizontal velocity. However, errors in both horizontal and angular measures were 2-fold higher for 30 than 90 degree camera angles. Our results agree with previous studies which reported that 90 degree camera angles provided the most accurate horizontal and angular outcomes for Kinovea analysis of body movements in hockey [51], simulated walking [44], and single leg squats [52].

We also examined how the accuracy of Kinovea outcomes differed between 1D versus 2D calibration approaches. We found that the accuracy of outcomes based on 2D calibration decreased with rotation of the calibration grid by 30 degrees or more, and by grid translation of 30 cm or more, indicating the value of accurate positioning of the grid near the plane of the fall. Our 1D calibration results, based on participant height, are valuable for researchers who

are unable to duplicate our approach to 2D video calibration, which requires revisiting the scene of the fall and capturing an image of a 1.6 x 1.6 m grid placed in the plane of the fall, with the same (unmoved) camera that recorded the fall. In general, there was little difference between 1D and 2D calibration in the accuracy of vertical outcomes, but errors in horizontal velocity were 1.6-fold higher for 1D than 2D calibration. However, for video collected from cameras oriented at 90 degrees, 1D calibration provided vertical, horizontal and angular velocity measures that were nearly as accurate (to within 4%) as 2D calibration values. Collectively, our results indicate that 1D video calibration based on height was nearly as accurate as 2D (grid) calibration for vertical measures, and for horizontal and angular measures when the camera was oriented at 90 degrees to the plane of the fall.

This study has important limitations. First, all of our data was collected from three healthy young adults, who experienced falls onto a firm gym mat, that were either self-initiated (simulating incorrect weight shifting), or caused by rapid translation of the ground surface. We see no reason why our approaches would not yield similar accuracy in estimating the kinematics of falls in different populations, including older adults. However, movement patterns during falls are diverse, and specific types of falls outside the range we examined (e.g., falls on stairs, falls from an initial sitting position, or falls involving excessive axial rotation during descent) may yield different accuracy from Kinovea. Second, when visible in the video footage, the reflective markers used for 3D motion capture helped in identifying body part locations to track in Kinovea. The lack of distinct landmarks on body parts in real-life falls may affect the accuracy of the measurements. Third, we only examined video footage having a resolution of 640 x 480 pixels, and our results may not reflect the accuracy that may be observed for video of lower or higher resolution. Additionally, we did not correct for camera lens distortion, which may have affected the accuracy of digitized body parts, particularly when they appeared at the edges of the video image, far from the calibration grid. Finally, studies are needed to compare the accuracy of Kinovea to alternative approaches for extracting kinematics from planar video (e.g., DeepLabCut [53]).

In summary, our results demonstrate that Kinovea can be successfully applied to extract whole-body kinematics from video footage of real-life falls, which is increasingly available [37]. However, lower accuracy was observed for angular kinematics of the upper and lower limb in sideways falls, and for horizontal measures from 30 degree cameras. 1D video calibration based on height was nearly as accurate as 2D (grid) calibration for vertical measures, and for horizontal and angular measures from 90 degree camera angles. The acceptability of the observed errors for a given user will depend on the study design and research questions. Our results can guide users in identifying the types of falls, camera angles, calibration techniques, and kinematic outcomes that meet a given desired level of accuracy in Kinovea.

## Supporting information

**S1 Table. Differences in positions and velocities between Qualisys ground truth signal and Kinovea signals.** The table shows differences between Qualisys ground truth signal (Q) and Kinovea signals (K) in (A) vertical, (B) horizontal and (C) angular position and velocity signals. Comparisons show the effects of different (i) cut-off frequencies, (ii) fall directions, (iii-vi) body parts for each fall direction, (vii) camera angles, and (viii-xvii) calibration techniques and orientations. In the camera angles comparisons, Qualisys values (vertical, horizontal, and angular) are different between the 90 degree, and the 60 and 30 degree cameras because markers from the right versus the left side of the body were analyzed, respectively. RMSE and NRMSE are based on comparing signals throughout the fall. Results show mean ± 1 SE values for all falls in a given category. For positions, values less than 0.001 m are rounded up. For

velocities, values less than 0.01 m/s are rounded up.
(DOCX)

**S2 Table. Raw unfiltered position data from Kinovea.** The table shows time-varying horizontal (X) and vertical (Z) positions in meters for each body part. Each sheet in the table is a separate fall.
(XLSX)

**S3 Table. Raw unfiltered position data from Qualisys during backward falls.** The table shows time-varying 3D positions (in meters) for all body parts during backward falls. The X axis represents horizontal position in the plane of the fall, Y represents horizontal position perpendicular to the plane of the fall, and Z represent vertical position. Each sheet in the table is a single fall. Frames corresponding to fall initiation and end of the fall are noted.
(XLSX)

**S4 Table. Raw unfiltered position data from Qualisys during forward falls.** The table shows time-varying 3D positions (in meters) for all body parts during forward falls. The X axis represents horizontal position in the plane of the fall, Y represents horizontal position perpendicular to the plane of the fall, and Z represent vertical position. Each sheet in the table is a single fall. Frames corresponding to fall initiation and end of the fall are noted.
(XLSX)

**S5 Table. Raw unfiltered position data from Qualisys during sideways falls.** The table shows time-varying 3D positions (in meters) for all body parts during sideways falls. The X axis represents horizontal position in the plane of the fall, Y represents horizontal position perpendicular to the plane of the fall, and Z represent vertical position. Each sheet in the table is a single fall. Frames corresponding to fall initiation and end of the fall are noted.
(XLSX)

**S6 Table. Results of the measures of accuracy between Qualisys ground truth signal and Kinovea signals.** The table shows measures of accuracy (RMSE, NRMSE and differences in peak velocities) for Kinovea results, used in statistical analyses of the effect of cut-off frequency, fall direction, camera angle, and calibration technique and orientations. Each type of comparison includes separate sheets in the table for vertical, horizontal, and angular measures.
(XLSX)

## Author Contributions

**Conceptualization:** Nataliya Shishov, Stephen N. Robinovitch.

**Data curation:** Nataliya Shishov, Karam Elabd, Vicki Komisar.

**Formal analysis:** Nataliya Shishov, Helen Chong.

**Investigation:** Nataliya Shishov, Helen Chong.

**Methodology:** Nataliya Shishov, Karam Elabd, Vicki Komisar, Helen Chong, Stephen N. Robinovitch.

**Project administration:** Nataliya Shishov.

**Resources:** Stephen N. Robinovitch.

**Visualization:** Nataliya Shishov.

**Writing – original draft:** Nataliya Shishov, Stephen N. Robinovitch.

**Writing – review & editing:** Nataliya Shishov, Karam Elabd, Vicki Komisar, Helen Chong, Stephen N. Robinovitch.

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
