## [Decision Letter · Decision Letter 0]

8 Jul 2021

PONE-D-21-13568

Accuracy of Kinovea software in estimating body segment movements during falls captured on standard video: effects of fall direction, camera perspective and video calibration technique

PLOS ONE

Dear Dr. Shishov,

Thank you for submitting your manuscript to PLOS ONE. After careful consideration, we feel that it has merit but does not fully meet PLOS ONE’s publication criteria as it currently stands. Therefore, we invite you to submit a revised version of the manuscript that addresses the points raised during the review process.

We look forward to receiving your revised manuscript.

Kind regards,

J. Lucas McKay, Ph.D., M.S.C.R.

Academic Editor

PLOS ONE

Journal Requirements:

3.  We note that Figure 1 includes an image of a participant in the study. 

Reviewers' comments:

Reviewer's Responses to Questions

**Comments to the Author**

1. Is the manuscript technically sound, and do the data support the conclusions?

Reviewer #1: Partly

Reviewer #2: Partly

2. Has the statistical analysis been performed appropriately and rigorously? 

Reviewer #1: Yes

Reviewer #2: Yes

3. Have the authors made all data underlying the findings in their manuscript fully available?

Reviewer #1: Yes

Reviewer #2: Yes

4. Is the manuscript presented in an intelligible fashion and written in standard English?

Reviewer #1: Yes

Reviewer #2: Yes

5. Review Comments to the Author

Reviewer #1: This study sought to determine the accuracy with which an open source biomechanical software package, Kinovea, could be used to estimate the kinematics of laboratory induced falls with respect to a gold standard 3D motion capture system. This is an interesting paper, which pending revision, may lower the bar for entry into the analysis of real-world falls. This is a big positive for those of us involved in falls research. The authors therefore are to be commended for taking on this project, and doing so with such breadth of independent variables. That being said, my enthusiasm for the manuscript was somewhat tempered by a variety of concerns regarding methodology, as well as presentation and interpretation of study results. These included but were not limited to the absence of any recommendations for the use of Kinovea when analyzing falls, complex figures (which may be better suited as tables given the volume of data), and the restatement of results within the discussion.

Major

1. It would be extremely helpful to the reader, and ease interpretation as well as future application, if the authors provided guidelines or recommendations on the use of Kinovea for the assessment of falls videos based on the observed RMSE values (i.e., when is the RMSE ok versus "too much"?). For example, should Kinovea be used for all fall directions? Both position and velocity data? All body segments? While this is hinted at in the conclusion, it is not well described with specifics in the discussion.

2. The authors describe difficulty with differentiation of position signals to velocity data. Did the authors consider or might the consider using a Svelsky-Golay filter? We have found this quite effective in calculating velocity signals from position data with minimal noise.

3. Figures 4 through 8 are a lot to take in. Given the density of the data would this data be better presented as tables?

4. Much of the discussion, and especially the first 2-3 sentences of nearly every paragraph, consists of a restatement of the results. This seems unnecessary. The discussion could be better focused on how and when to use Kinovea for the analysis of falls based on comment 1 above. This may align the discussion with the results quite well.

5. Could the authors provide a justification for the use of Kinovea versus some other open source software package like DeepLabCut that has become popular for the analysis of video data. Is a comparison of Kinovea to DeepLabCut necessary?

Minor

1. Could the authors clarify why they chose to calculate segment rather than anatomical joint angles?

2. Could the authors clarify whether reflective markers used in the 3D analysis were or were not used when processing the Kinovea data? If so, what implications might this have for the application of Kinovea to video without passive reflective markers?

3. In the methods the authors describe an assessment of inter-rater reliability using RMSE. It is not immediately clear if this is an appropriate method by which to assess inter-rater reliability. Should the authors be determining inter-rater reliability using intra-class correlation coefficients (ICC)? The following citation may provide guidance in this decision.

Weir JP. Quantifying test-retest reliability using the intraclass correlation coefficient and the SEM. J Strength Cond Res. 2005;19(1):231-240.

Reviewer #2: Accuracy of Kinovea software in estimating body segment movements during falls captured on standard video: effects of fall direction, camera perspective and video calibration technique

Summary of reviewer comments: The manuscript focuses on a significant and clinically-relevant problem –measurement of movement quality during falls in real-world settings. The study methods comprised laboratory-based falling experiments, and comparison of full body kinematics (linear and angular positions and velocities) measured from 3D motion capture to those estimated by Kinovea 2D digitization software from standard surveillance video cameras. The effects of several methodological factors on the accuracy of Kinovea were evaluated, including fall direction, camera angle, filtering cut-off frequency, and calibration techniques. The results are comprehensive and presented clearly. However, the rationale for the study and comparison of current work in light of previous literature needs strengthening. The study is limited by a very small sample size (N=3), which may be caused by the unique methodology employed but needs justification. The discussion can benefit from reorganization and editing.

Specific comments are listed below:

Abstract – The statements regarding the results can benefit from inclusion of some numbers, e.g. “When compared to 90 deg, a 60 deg camera angle yielded less accurate horizontal velocities (include magnitude or % decrease in error here). The conclusion of the abstract could be edited to be more specific and to better discuss the implications of the results.

Introduction –

“Improved understanding of the kinematics of real-life falls should help to inform efforts for injury prevention” – it would strengthen the introduction if the authors could provide details or examples of how measuring kinematics during falls can inform fall prevention programs.

“compared to 3D motion capture, Kinovea estimated hip, knee, and ankle angles during walking with errors less than 5 deg” – please clarify if this stated accuracy or error is for 2-D or sagittal plane angles? Also, please refrain from using “deg” instead of degrees throughout the text of the manuscript.

Because the manuscript has several objectives related to evaluation of Kinovea for measurement of kinematics during falls, it would improve clarity if the authors could edit the last para of the Introduction to clearly list the study objectives (in order of importance), and also provide the rationale or importance of each stated objective. Additionally, if possible, a couple sentences about previous studies that may have addressed the same or similar questions would be beneficial for the reader.

Methods –

The study involves a very small sample size (3 individuals). Please clarify the rationale for this sample size in the methods section, and a statement about the sample size as a limitation in the Discussion section.

The methods used for the 2 different camera calibration methods should be described in more detail, and appropriate supportive references provided, if available.

There may be a formatting error in Table 1 (columns on the far right appear truncated/cropped).

“We used one-way ANOVAs to examine how the accuracy of Kinovea signals associated with Kinovea filter cut-off frequency, camera angle, fall direction, body part, and calibration technique”- this description is somewhat confusing as written. Was the ANOVA used to test associations or the effect of these methodological variables on errors? Were separate ANOVAs performed for each variable (e.g. camera angle, filter settings, etc)? Please edit to improve clarity. Similarly, the rationale for repeated measures comparisons for the same fall could be better clarified.

Results and Discussion –

The Discussion can be strengthened by including more detailed interpretation of the study findings, placing the current results in context of previous literature, as well as methodological and clinical implications or insights gained from the study.

The discussion should also include potential reasons for instances where fairly high errors were observed (e.g. Figure 7 C shows high angular position errors for certain segments).

Figure 1 could benefit from additional labels to show the relevant objects / landmarks in each panel.

6. PLOS authors have the option to publish the peer review history of their article (what does this mean?). If published, this will include your full peer review and any attached files.

Reviewer #1: No

Reviewer #2: No

---

## [Author Response · Author response to Decision Letter 0]

4 Sep 2021

As part of this submission, attached are a Cover Letter to the Academic Editor, and Response to Reviewers document in which specific reviewer and editor comments are addressed.

---

## [Decision Letter · Decision Letter 1]

11 Oct 2021

Accuracy of Kinovea software in estimating body segment movements during falls captured on standard video: effects of fall direction, camera perspective and video calibration technique

PONE-D-21-13568R1

Dear Dr. Shishov,

We’re pleased to inform you that your manuscript has been judged scientifically suitable for publication and will be formally accepted for publication once it meets all outstanding technical requirements.

Kind regards,

Manabu Sakakibara, Ph.D.

Academic Editor

PLOS ONE

Additional Editor Comments (optional):

Reviewers' comments:

Reviewer's Responses to Questions

**Comments to the Author**

1. If the authors have adequately addressed your comments raised in a previous round of review and you feel that this manuscript is now acceptable for publication, you may indicate that here to bypass the “Comments to the Author” section, enter your conflict of interest statement in the “Confidential to Editor” section, and submit your "Accept" recommendation.

Reviewer #1: All comments have been addressed

2. Is the manuscript technically sound, and do the data support the conclusions?

Reviewer #1: Yes

3. Has the statistical analysis been performed appropriately and rigorously? 

Reviewer #1: Yes

4. Have the authors made all data underlying the findings in their manuscript fully available?

Reviewer #1: Yes

5. Is the manuscript presented in an intelligible fashion and written in standard English?

Reviewer #1: Yes

6. Review Comments to the Author

Reviewer #1: (No Response)

7. PLOS authors have the option to publish the peer review history of their article (what does this mean?). If published, this will include your full peer review and any attached files.

Reviewer #1: No

---

## [Editor Report · Acceptance letter]

14 Oct 2021

PONE-D-21-13568R1 

Accuracy of Kinovea software in estimating body segment movements during falls captured on standard video: effects of fall direction, camera perspective and video calibration technique 

Dear Dr. Shishov:

I'm pleased to inform you that your manuscript has been deemed suitable for publication in PLOS ONE. Congratulations! Your manuscript is now with our production department. 

Kind regards, 

on behalf of

Dr. Manabu Sakakibara 

Academic Editor

PLOS ONE